

# Probing many-body localization in a disordered quantum dimer model on the honeycomb lattice

Francesca Pietracaprina[1,2⋆] and Fabien Alet[1†]

**1** Laboratoire de Physique Théorique, IRSAMC, Université de Toulouse, CNRS, UPS, France
**2** School of Physics, Trinity College Dublin, College Green, Dublin 2, Ireland

⋆ pietracaprina@irsamc.ups-tlse.fr † alet@irsamc.ups-tlse.fr

## Abstract

We numerically study the possibility of many-body localization transition in a disordered quantum dimer model on the honeycomb lattice. By using the peculiar constraints of this model and state-of-the-art exact diagonalization and time evolution methods, we probe both eigenstates and dynamical properties and conclude on the existence of a localization transition, on the available time and length scales (system sizes of up to $N = 108$ sites). We critically discuss these results and their implications.



# 1   Introduction

Localization in disordered, interacting quantum systems [1, 2] is a topic that has recently received wide attention due to the very peculiar phenomenology [3–6], the foundational issues about quantum integrability and ergodicity involved [7, 8], and the increased precision and control on experimental realizations [9, 10]. Systems with a many-body localization (MBL) transition typically exhibit two regimes, one at low disorder which obeys the eigenstate thermalization hypothesis (ETH) and one at high disorder which exhibits no transport, no thermalization [11–14] and emergent integrability due to an extensive number of quasi-local integrals of motion [15–19]. Furthermore, localized states have low entanglement at any energy and obey an area law, a property usually valid for ground states only [20, 21]. Finally, localization in interacting systems is characterized by the very slow spreading of information, namely the entanglement [22–24], and the total absence of transport for local observables [1, 2]. All these features have contributed to make MBL a compelling physical phenomenon, including with respect to quantum information processing protocols [20, 25–27].

In the context of the study of MBL transitions, a wide range of results, outlining the phenomenology described above, have been produced for one-dimensional (1D) systems [3–6]. Remarkably, a proof of the existence of the MBL transition has been obtained for a 1D quantum Ising model with a transverse field [28, 29]. In higher dimensions, however, no such proof exists. One generally expects that in higher dimensions delocalization is favoured due to the increase in channels for the delocalizing terms, similarly to the phenomenology of Anderson localization in higher dimensions. More specifically, general arguments based on the existence and size-scaling of thermalizing bubbles support the absence of localization for large enough times [30, 31], even though no rigorous proof was obtained either.

A number of results on 2D systems have notably been presented. Experimental results obtained in cold atoms setups interestingly show absence of dynamics and localization at high disorder [10, 32]. At present, this experimental evidence is arguably of higher quality than the analytical and numerical modeling of MBL in 2D. Numerically, a number of approaches have been explored in 2D lattice models, using both unbiased (not involving approximations or assumptions about the physical properties of the underlying system) and biased methods, and showing indications of a localized regime [33–38]. Other simulations conclude in favor of absence of MBL [39]. However, the main limit of numerical approaches is the small system sizes and/or time scales that are reachable in the computations. The size of the Hilbert space and thus of the quantum problem grows exponentially with the number of particles $N$ in the system while the physical lengthscale of the sample grows as a square root of $N$. For unbiased methods this is an especially strong constraint, effectively limiting the analysis to systems up to around 20 spins-1/2. While in one dimension several different lattice sizes can fulfill this requirement, thus allowing in principle finite size scaling to be performed, this is no longer the case in two dimensions where the number of system sizes are greatly limited. While larger system sizes can be reached using methods geared towards capturing properties of an MBL regime [33–35, 40–42], these methods are not unbiased and by construction will miss the ergodic regime or the phase transition.

Here, we aim to investigate an MBL transition in a specific system up to a real-space size as large as possible and with unbiased methods. We do this by considering a highly constrained model and state-of-the-art numerically exact methods [43]. Specifically we consider a disordered quantum dimer model (QDM) on a honeycomb lattice, where each lattice link is either free or occupied by a dimer with the constraint that each lattice site is touched by one and only one dimer [44–46]. An immediate consequence for this is that the dynamics of such a model is very constrained: single-dimer moves are not allowed and the simplest move involves an hexagonal plaquette. Moreover, this constraint also automatically encodes strong interac-

tions which for the honeycomb lattice already imply long-range correlations in the statistical ensemble of dimer coverings. The interplay between a constrained dynamics, which favors slow dynamics and localization [47], and the strong interactions, which favor delocalization, creates an ideal situation for an MBL transition to exist. Finally, we note that such models are based on Hilbert spaces that, due to the constraints, have considerably lower dimension compared to spin systems: for $N$ 1/2-spins, the Hilbert space size is $2^N$, while it scales only as $\approx 1.175^N$ [44] for a dimer system on a $N$-sites honeycomb lattice, giving an obvious numerical advantage for large system sizes. A previous work has analyzed a similar disordered QDM on a square lattice [48]. Here, we substantially push forward this analysis, almost doubling the maximum system size reached, by turning to the honeycomb lattice instead.

The article is structured as follows. In Section 2 we detail the model Hamiltonian, the symmetry sectors and the lattices used as well as the procedures used to obtain the numerical results. Such results are outlined in Section 3, first considering observables within exact mid-spectrum eigenstates, and, secondly, the dynamical properties obtained with Krylov time evolution. Finally, we provide conclusions in Section 4. In the appendix we discuss in detail the lattice clusters used in the numerical analysis (Appendix A), further energy-resolved quantities (Appendix B) and comparisons with the entanglement properties of specific states (Appendix C).

## 2  Model

We consider the following quantum dimer model on the honeycomb lattice [45, 46, 49] with a random potential:

$$H_{\mathrm{QDM}} = -\tau \sum_{\bigcirc_p} \left( |\hexflip\rangle\langle\hexflip| + |\hexflip\rangle\langle\hexflip| \right) + \sum_{\bigcirc_p} v_p \left( |\hexflip\rangle\langle\hexflip| + |\hexflip\rangle\langle\hexflip| \right) \qquad (1)$$

The first term, an hexagon "flip", is a kinetic term. The second term is a disordered potential on each flippable hexagon; the $v_p$ are drawn from a uniform distribution in $[-V, V]$.

We construct lattices with $N = 42, 54, 72, 78, 96$ and $108$ sites; in Fig. 1 (a) we show the $N = 72$ lattice and we refer the reader to the Appendix for more details on the other clusters. On the honeycomb lattice with periodic boundary conditions, the constraints due to the dimers and to the allowed plaquette moves are such that two conserved quantities, the winding numbers, exist. The winding numbers are defined as the sum along a line parallel to the $x$ or $y$ axis; having labeled the honeycomb lattice sites with binary alternating symbols $A$ (yellow in Fig. 1) and $B$ (green) and orienting all links from $A \rightarrow B$, we add a +1 value to the sum if the line crosses a dimer with an arrow in the positive direction, −1 if the arrow is in the negative direction and 0 if there is no dimer (see Fig. 1 (c)). Among the sectors with conserved total winding number, we select the one for which $w_x = w_y = 0$, which is the largest one. We remark that, for finite lattices, not all lattice shapes allow the existence of this zero winding sector; we discard lattice shapes that do not satisfy this requirement [50].

Table 1 displays the number of allowed coverings in the zero winding sector, which corresponds to the size $\mathcal{N}_H$ of the Hilbert space. The number of nonzero elements in the matrix is also noted, which, in addition to matrix size, contributes to limiting the feasibility of the numerical calculations.

We perform exact diagonalization on some of these lattices (up to size 78). We use either full diagonalization or shift-invert methods [43] to obtain around 100 eigenstates at the center of the spectrum. We also study the dynamics of nonequilibrium initial states though Krylov subspace time evolution methods for all lattice sizes [51]. In all cases, we average over disorder

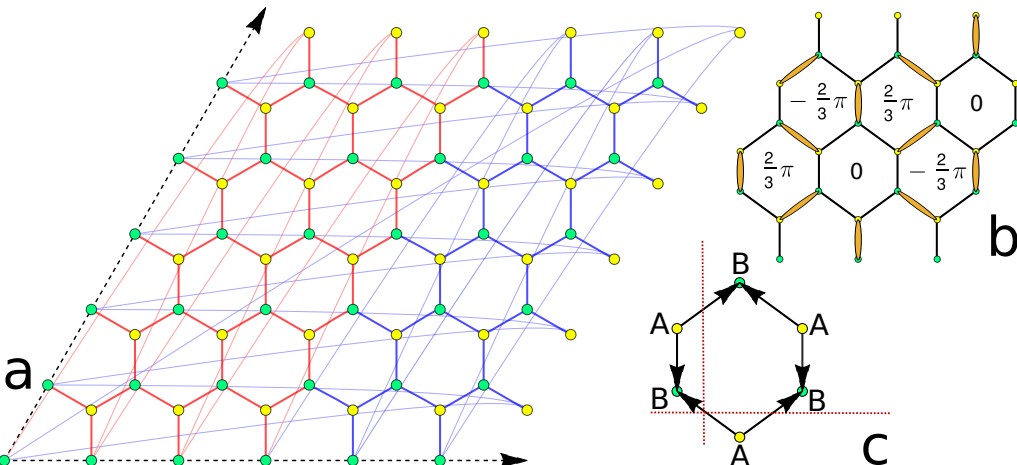

Figure 1: Dimers on the honeycomb lattice. (a) Untilted $N = 72$ cluster. We consider periodic boundary conditions. The lattice is split in two parts (here shown in red and blue) for the computation of the bipartite entanglement. (b) Example of dimer covering, namely the star configuration that is used as a reference for the computation of the imbalance (Sec. 3.2) and as initial state for the quenched dynamics (Sec. 3.4). The definition of the phases $\phi_p$ used to define the imbalance in Sec. 3.2 are shown for each plaquette. (c) Computation of the two independent winding numbers on a single plaquette.

Table 1: Matrix size $\mathcal{N}_H$ and number of nonzero elements *nnz* for the clusters that have been considered.

| Cluster size $N$ | Coverings $w_x = w_y = 0$ $\mathcal{N}_H$ | Nonzero elements *nnz* |
|---|---|---|
| 42 | 1 032 | 8 046 |
| 54 | 7 311 | 69 519 |
| 72 | 131 727 | 1 596 927 |
| 78 | 349 326 | 4 536 288 |
| 96 | 6 460 809 | 100 676 169 |
| 108 | 45 649 431 | 791 275 167 |

realizations of the random potential (at least 1000 for most system sizes and around 100 for the dynamics on the largest one).

# 3 Results

We consider various quantities with known different behaviors in the MBL and ETH regimes. We analyze spectral, eigenstate, and entanglement properties as well as the dynamics of the system.

## 3.1 Spectral properties

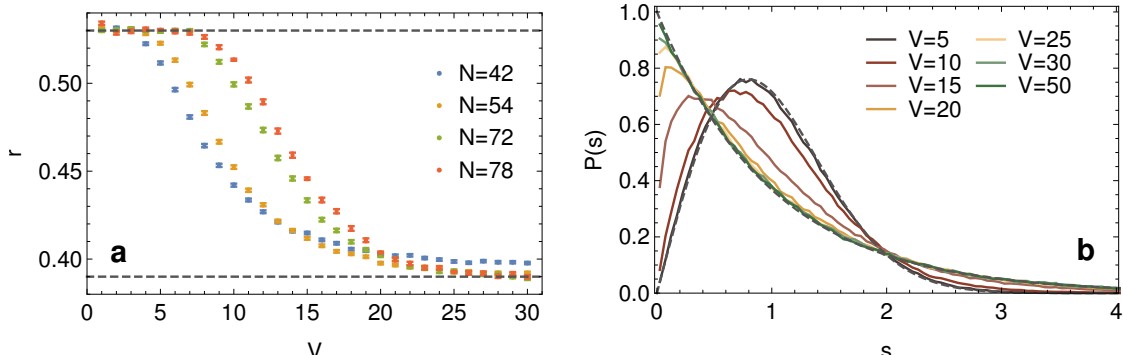

Figure 2: *Panel a*. Gap ratio $r$ as a function of the disorder strength $V$ for different honeycomb samples. The two limiting values at approximately 0.53 and 0.39 (dashed lines) correspond to the ones obtained for a Wigner-Dyson and Poisson distribution respectively. *Panel b*. Distribution of the energy gaps $s_i$ of the unfolded spectrum. The Wigner-Dyson and Poisson reference distributions are shown in dashed lines.

**Spectral gap ratio**     We start by analyzing the spectral properties of the two regimes. Specifically, we consider the energy level gap ratio [14]:

$$\langle r \rangle = \left\langle \frac{\min(s_i, s_{i+1})}{\max(s_i, s_{i+1})} \right\rangle_i, \tag{2}$$

where $s_i = E_{i+1} - E_i$ is the gap between two adjacent eigenvalues. We average in a small window of about 100 eigenstates around the center of the spectrum as well as over disorder realizations. Depending on the level gap statistics, $\langle r \rangle \approx 0.39$ for a Poisson distribution in the localized phase and $\langle r \rangle \approx 0.53$ [52] for a Wigner-Dyson distribution corresponding to the ETH phase.

In Fig. 2, panel **a**, we show the value of $\langle r \rangle$ as a function of the disorder for various system sizes. It appears that both localized and ETH regimes are captured with the available cluster sizes. The transition value can typically be inferred by where the curves for increasing size cross, as it denotes opposite flows in the system size scaling in the two regimes. We note that here the crossing point has a noticeable drift towards higher $V$ values.

In the panel **b** of Fig. 2 we show the probability distributions of the gaps $s$ of the unfolded spectrum for various values of the disorder $V$, showing excellent agreement with a Poissonian or a Wigner-Dyson distribution (shown in black) for high and low $V$ respectively.

For the smallest sizes $N = 42$ and $N = 54$, we additionally computed the gap ratio as a function of the energy density (not just for the middle of the spectrum), see Appendix B.

## 3.2   Eigenstates

**Kullback-Leibler divergence for energy-adjacent eigenstates**     We now consider quantities characterizing eigenstate properties which have been shown to be good indicators of localization. In the localized phase, eigenstates and local observables close in energy are very different in structure, as opposed to the ETH phase. Thus, we consider the Kullback-Leibler divergence for two consecutive eigenstates $|\psi\rangle$ and $|\psi'\rangle$ in the spectrum, defined as

$$\mathrm{KL} = \sum_i |\langle \psi | b_i \rangle|^2 \ln \frac{|\langle \psi | b_i \rangle|^2}{|\langle \psi' | b_i \rangle|^2}, \tag{3}$$

where the sum runs over the $\mathcal{N}_H$ elements $|b_i\rangle$ of the Hilbert space basis. We expect KL to approach to $\text{KL}_{\text{GOE}} = 2$ (the value obtained for the Gaussian orthogonal ensemble of random matrices) in an ETH regime and to diverge with system size in a localized regime [12].

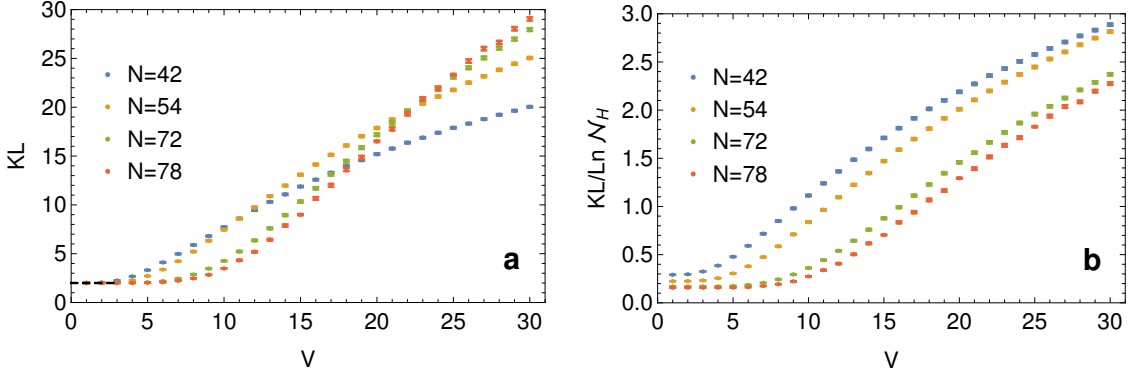

Figure 3: *Panel a.* Kullback-Leibler divergence KL of eigenstates adjacent in the many-body spectrum as a function of the disorder strength $V$ and for different samples. *Panel b.* Kullback-Leibler divergence rescaled by the system size.

We show the results for KL in Fig. 3a as a function of the disorder strength $V$. The limit value KL $= 2$ is well captured at small disorders $V$, as well as a crossing point between the $N = 54$, $N = 72$ and $N = 78$ clusters ($N = 42$ appears to show stronger deviations due to the small size), with some drift due to finite-size scaling, suggesting a localization transition around $V \approx 22 - 25$. In Fig. 3b we show KL rescaled by the system size. Although for the chosen range of disorder $V$ the curves cannot be seen to all collapse as expected, they do collapse in pairs (72 and 78, 42 and 58).

**Eigenstate participation entropy**    In a similar manner, we consider the participation entropy of the eigenstates, which gives information about localization in the Hilbert space [12, 53]. It is defined as

$$S_p = -\sum_i |\langle\psi|b_i\rangle|^2 \ln|\langle\psi|b_i\rangle|^2. \tag{4}$$

For a state which is localized in the Hilbert space, $S_p$ is of $O(1)$. For many-body localized states, a multifractal behavior is expected in this computational basis [53], with a participation entropy behaving as $S_p \propto a \ln\mathcal{N}_H$, with $a < 1$. For extended states in the ETH regime, $S_p$ will scale as $\ln\mathcal{N}_H$, with $a = 1$.

In Fig. 4 we show the participation entropy, rescaled by $\ln\mathcal{N}_H$ (i.e. this ratio is the coefficient $a$ up to higher order corrections), as a function of the disorder $V$. At low disorder we see that $a$ has a high value which is likely to scale to 1 with increasing size. A different behavior onsets at around $V \approx 20 - 25$: the curves for different system sizes join and collapse, suggesting a finite $a < 1$ asymptotically for disorders larger than this value.

**Eigenstate imbalance**    We next consider the imbalance of the eigenstates with respect to a specific configuration where the state used as a reference is chosen as the basis element of the so-called star configuration displayed in Fig. 1b. We define the (complex) imbalance as

$$\mathcal{I} = \frac{1}{N_p}\sum_{\bigcirc_p} e^{i\phi_p}\left(|\hexagon\rangle\langle\hexagon| + |\hexagon\rangle\langle\hexagon|\right). \tag{5}$$

The phases $\phi_p$ assume three possible values: $0$, $\frac{2}{3}\pi$ and $-\frac{2}{3}\pi$, depending on the dimer configuration on the plaquette $p$ (see Fig. 1b). With this definition, the imbalance of the reference

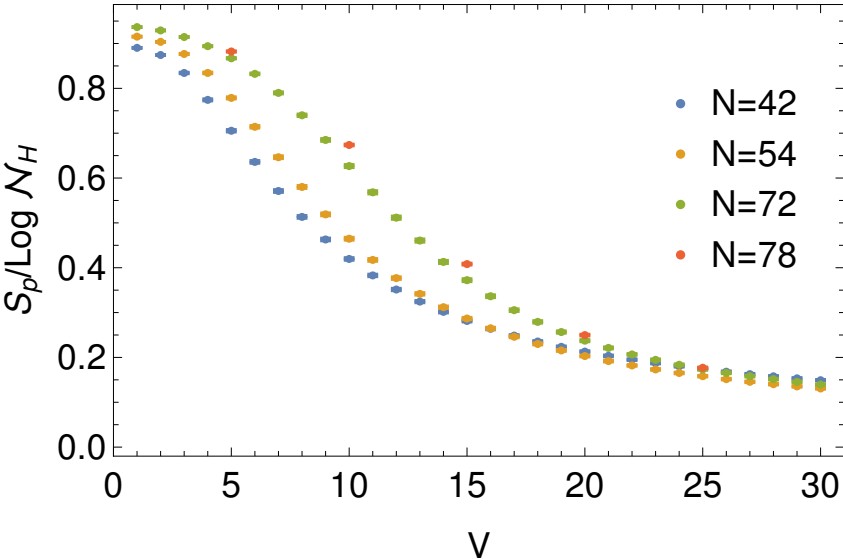

Figure 4: Eigenstates participation entropy rescaled by the logarithm of the Hilbert space size, $S_p/\ln\mathcal{N}_H$, as a function of disorder strength $V$ and for different samples.

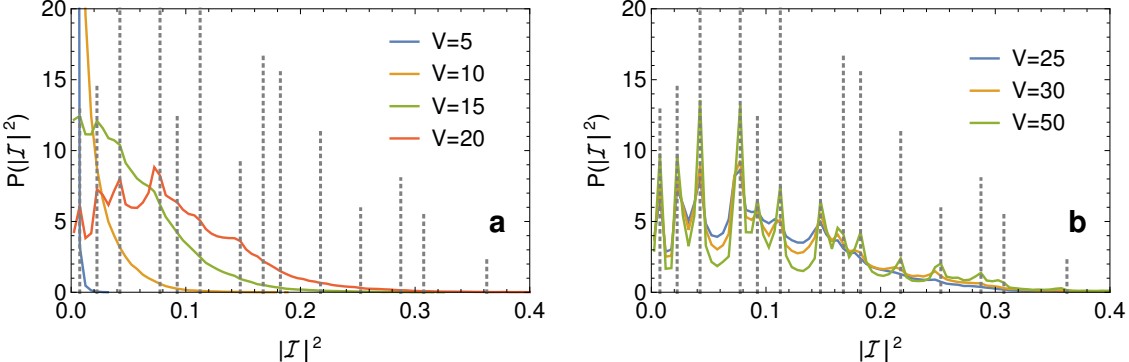

Figure 5: Probability distribution of modulus-squared imbalance for system size $N = 78$. The imbalance of the configuration basis states are shown as dashed lines. *Panel a.* Small disorders ($V = 5$ to 20) show a distribution peaked in 0 for small disorder ($V = 5, 10$) and broadening and development of peaks corresponding to the imbalance of basis configurations (for $V = 20$). *Panel b.* Large disorders display a clear structure with peaks located at the imbalance of basis states, while maintaining a continuous distribution.

basis state in Fig. 1b has $\Re I = 1$ and maximum amplitude $|I|^2 = 1$. Delocalized eigenstates will have a probability distribution for the modulus squared imbalance which is sharply peaked in 0. On the other hand, if states are localized and close to basis states, the imbalance will be peaked around the values corresponding to dimer configurations of the basis states.

The probability distribution of the modulus squared of the imbalance is shown in the two panels of Fig. 5 respectively for low (panel **a**) and high (panel **b**) values of disorder for the largest system size ($N = 78$). As expected, the imbalance distribution is sharply peaked at 0 for very small values of disorder with an increasing variance for higher disorders. Between $V = 15$ and $V = 20$ the distribution broadens and develops peaks at values $|\mathcal{I}|^2 > 0$ which, at higher disorder ($V \geq 25$), are shown to closely correspond to the distribution of the imbalance of the configuration basis states (shown in dashed lines in Fig. 5), which become eigenstates in the infinite-disorder limit.

As for 1D systems, the imbalance is a quantity especially useful for characterizing the dynamical properties of the system in different regimes. We will further analyze dynamics of the imbalance after a quench in Sec. 3.4.

**Eigenstate dimer bond occupation** We finally consider a local observable, the dimer bond occupation, and specifically the probability distribution of

$$O_k = \langle \psi | n_k | \psi \rangle, \tag{6}$$

where the operator $n_k$ acts on the basis vectors $|b_i\rangle$ as $n_k |b_i\rangle = 1$ if bond $k$ is occupied in $b_i$ and 0 otherwise.

In the limit of a uniformly extended state, all three bonds belonging to a site have the same probability to be occupied, i.e. $1/3$. As shown in the main panel of Fig. 6, for a delocalized eigenstate, this translates into a probability distribution of $O$ sharply peaked at $1/3$ for low disorder values, with an increasing variance for higher disorders. At disorder between $V = 15$ to $V = 20$ the distribution becomes bimodal with two peaks at $O = 0$ and $O = 1$, meaning that the eigenstates start to resemble some given dimer configurations. In the limit of infinite disorder, where the eigenstates coincide with the configuration states, the distribution is $2/3\,\delta(0) + 1/3\,\delta(1)$, given that one bond per lattice site is occupied. In the inset of Fig. 6 the expected behavior is further evidenced by the computation of the integral of the peaks in small intervals near 0 (solid lines) and 1 (dashed lines) respectively; for increasing system size and disorder strength, the peaks approach $2/3$ and $1/3$ respectively.

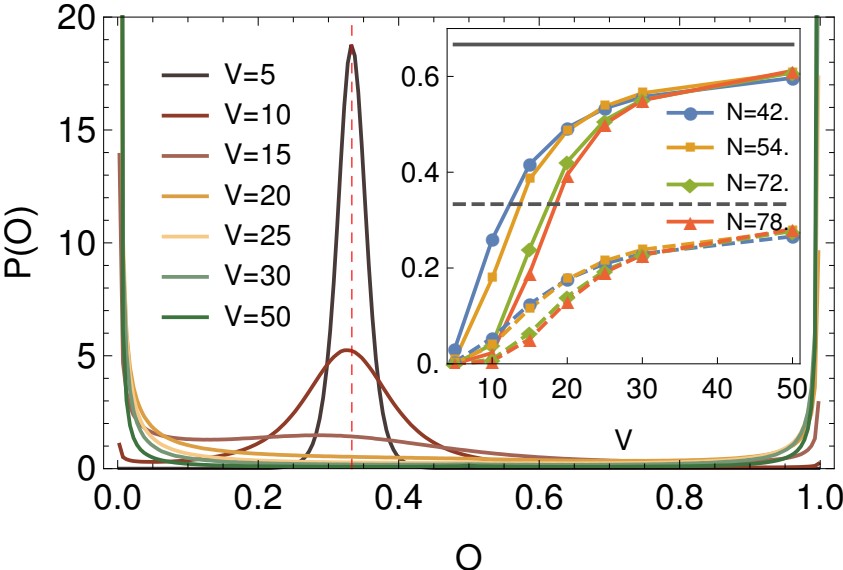

Figure 6: Probability distribution of bond occupation for the system size $N = 78$. *Inset.* Integral of $P(O)$ over a small interval ($\leq 5\%$) near $O = 0$ (solid lines) and $O = 1$ (dashed lines), compared to the respective limit values $2/3$ and $1/3$.

## 3.3 Half-system entanglement entropy

Next, we consider the entanglement properties of eigenstates through their von Neumann entanglement entropy

$$S = -\mathrm{Tr}\,\rho_A \ln \rho_A, \tag{7}$$

where $A$ is a region comprising half of the sample and $\rho_A = \mathrm{Tr}_B\,\rho$ is the reduced density matrix obtained from an eigenstate by tracing out the complementary region $B$. The analysis of the

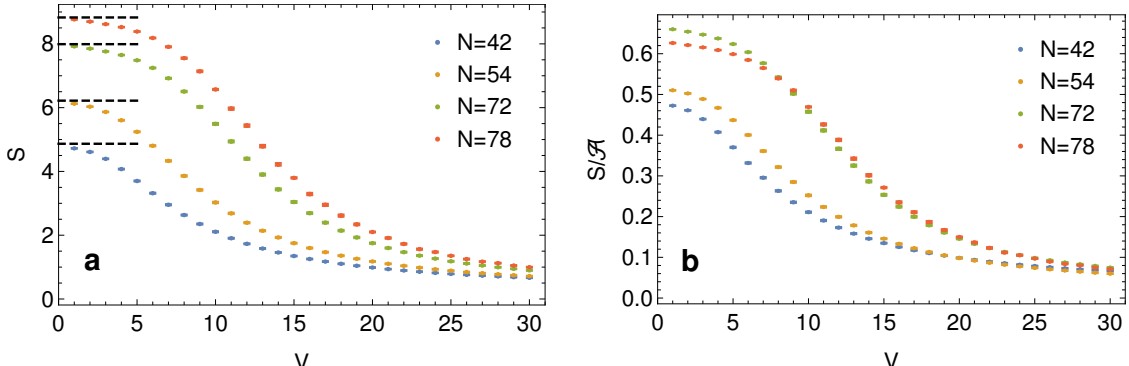

Figure 7: *Panel a*. Entanglement entropy $S$ as a function of the disorder $V$ and for different sample sizes. The dashed lines are the bipartite entanglement entropy for random states, averaged over $10^4$ realizations, and represent the limiting value of the entanglement entropy of eigenstates as $V \to 0$. *Panel b*. Entanglement entropy rescaled with the size of the boundary $\mathcal{A}$ of the bipartition, showing a collapse, and thus an area-law scaling, at high disorder values.

entanglement entropy has been especially useful in the study of MBL transitions given the low, area law entanglement of all localized states, to be compared with a volume law scaling in the extended regime [12, 20, 54, 55]. In the clusters taken into consideration, there is some freedom in the choice of the two regions $A$ and $B$; here, where possible, we consider a cut that runs parallel to the lattice vectors. The two regions are shown in red and blue respectively in Fig. 1 of the main text and in Fig. 14 in the appendix.

In the panel **a** of Fig. 7 we show the entanglement entropy Eq.(7) as a function of the disorder strength $V$ for different sample sizes. For low-$V$ values, we see that $S$ approaches the value obtained for random states (shown in the figure as a dashed line) as $V \to 0$, thus making evident a volume law entanglement. At high disorder, on the other hand, we observe an area law growth; specifically, by considering $S/\mathcal{A}$ where $\mathcal{A}$ is the length of the boundary between the two subsections, we observe a collapse (see Fig. 7 panel **b**). Interestingly, as seen for other quantities, the curves for different system sizes collapse in pairs, at around $V = 18$ for sizes $N = 42$ and $N = 54$ and at $V = 20$ for sizes $N = 72$ and $N = 78$, with both sets of curves collapsing only for larger $V$.

Given the relatively arbitrary choice of the boundary of the bipartition, as an additional comparison and justification for adequateness of the use of volume and $\mathcal{A}$ area laws, we considered the entanglement entropy of some special states. One class is the already mentioned random states with volume law entanglement growth. We additionally considered the uniform ('Rokshar-Kivelson' [45]) state, defined as $\psi_{\mathrm{RK}} = 1/\sqrt{\mathcal{N}_H} |1\,1\dots 1\rangle$ in the dimer covering configurations basis, as well as the ground state of the model (1) with no disorder and a small constant field $V_c$, which both have an area law entanglement scaling. The entanglement entropy computed for the clusters and the cuts under consideration indeed scales with $\mathcal{A}$ as expected (see Appendix C and Fig. 16).

In order to better understand the position of a transition point, we consider the variance of the entanglement entropy distribution as a function of disorder. The variance is expected to have a peak at the transition value (with possibly strong finite-size corrections) [54,55]. In the main panel of Fig. 8 we show the standard deviation $\sigma_S$ of $S$ for the eigenstates in the energy window around $E = 0$ and for different disorder realizations. A peak is present, although with a substantial drift towards higher disorder values. We attempted in two ways to understand the scaling of the peak with system sizes. Since there is no finite-size scaling theory associated to the putative MBL transition, we tried heuristically to plot the position of the peak rescaled

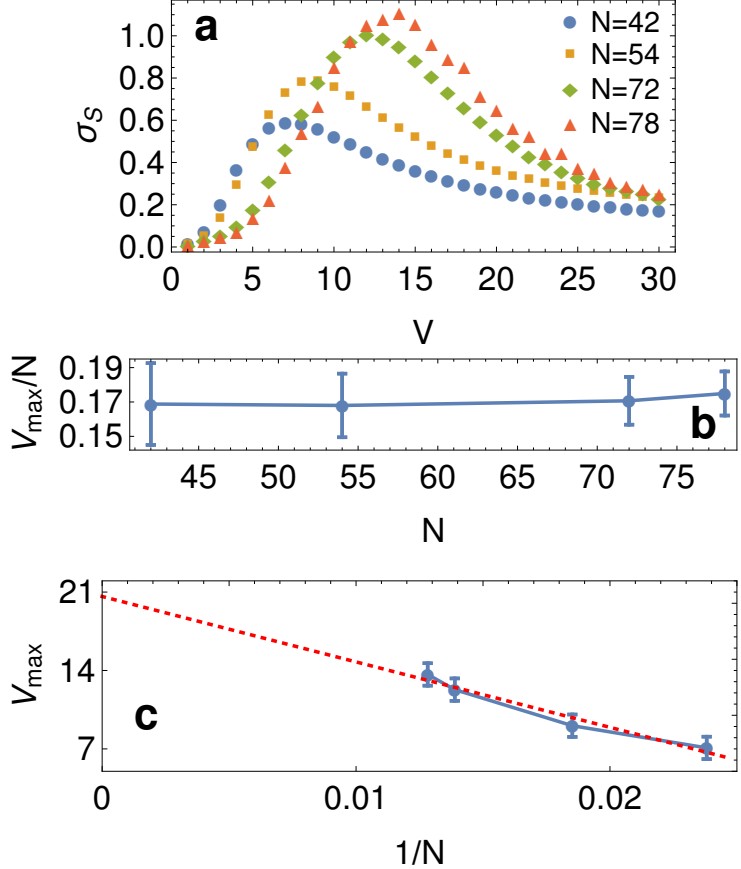

Figure 8: *Panel a*. Standard deviation of the entanglement entropy as a function of disorder for different sample sizes. *Panel b*. Position of the maximum entanglement entropy standard deviation $V_{max}$ scaled by the system size $N$, as a function of the system size. For the range of accessible sizes, $V_{max}$ has an approximately linear increase with system size. *Panel c*. A different attempt at scaling the position of the maximum, by plotting as a function of the inverse of the system size $1/N$. We show a linear fit as a dotted red line, which extrapolated to 0 would give an estimated transition at $V = 20 \pm 2$ (see the main text for a discussion on the quality of this extrapolation).

with cluster size (panel **b**) as well as $1/N$ extrapolation of this position (panel **c**). While the former seems to suggest that for the entanglement entropy, system sizes up to $N = 78$ do not show convergence to a finite transition value, we cannot exclude from the latter plot that it extrapolates to a finite value (e.g. using a $1/N$ fit). We note that it is especially difficult to be definitive in this extrapolation given that the position of the peak is only known within the precision of our disorder grid. Even being quite dense, a small change (by $\Delta V = 1$) of the position of the peak for the larger cluster can considerably change the extrapolation. We conclude that, solely based on the study of the peak of the standard deviation of entanglement entropy, we cannot predict whether the system sizes that we considered are still within the non-universal scaling regime or whether the transition does not hold asymptotically in the thermodynamic limit.

## 3.4 Dynamics

We finally consider the dynamical properties of the system. Starting from a product state, which is taken as an element of the computational basis, we perform a quench to the disordered model:

$$|\psi(t)\rangle = \exp(-iHt)|\psi(0)\rangle . \tag{8}$$

The chosen initial state $|\psi(0)\rangle$ is the same as the reference state for the imbalance calculation in Sec. 3.2. In the 1D MBL regime, transport of local quantities is absent and entanglement has a well-understood slow logarithmic growth [22, 56, 57]. We look for these markers of localization in the present model at high disoder. We consider the same clusters that have been used in the exact diagonalization analysis, that is $N = 42$, 54, 72 and 78, with the addition of the $N = 96$ and 108 clusters. The time evolution is performed through full exact diagonalization for the clusters $N = 42$ and 54, and with the Krylov method for the larger ones. We average over $10^4 \div 10^3$ disorder realizations for clusters up to $N = 96$ and around 100 realizations for the largest cluster $N = 108$.

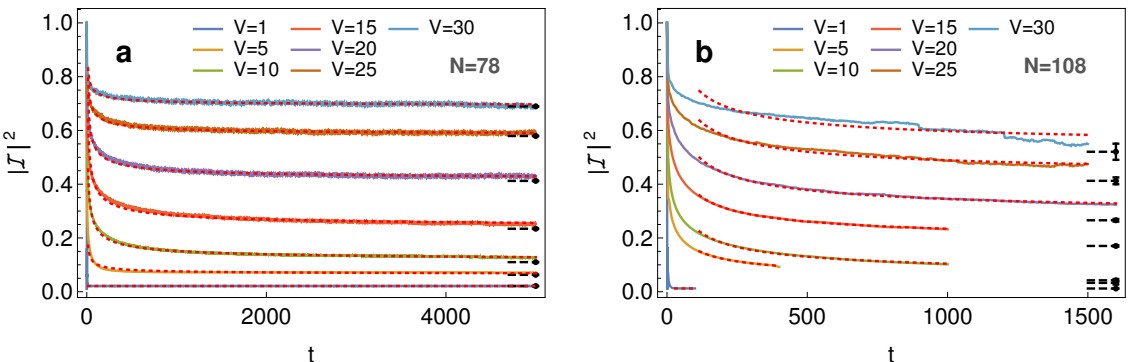

Figure 9: Modulus-squared imbalance $|I|^2$ as a function of time for the two clusters $N = 78$ (left) and $N = 108$ (right). The red dotted lines are $1/\sqrt{t}$ fits of the long time dynamics. The rightmost dashed lines and corresponding error bars are the extrapolated asymptotic values, also shown in Fig. 10.

**Imbalance** We start by considering the imbalance of the time-evolved state with respect to the initial state, as defined in Sec. 3.2 and Eq. (5). In Fig. 9 we show the modulus-squared imbalance as a function of time for various values of the disorder $V$ for the system sizes $N = 78$ and $N = 108$. An imbalance value $|I|^2 > 0$ indicates that some memory of the initial state is kept after the time evolution. For the smallest clusters $N = 42$ and $N = 54$ we are able to obtain the evolved states at very large times, for the larger sizes and with the Krylov time evolution we are only able to reach times of order $t = 1000$ (in units of the inverse of the plaquette flip energy scale $\tau$) according to the system size and the disorder strength. From Fig. 9, we see a decrease and, for most disorder values, a saturation of the imbalance (namely for the $N = 78$ cluster for which longer times are available).

We also attempt to estimate the asymptotic value, using a fit of the form $\sim t^{-1/2}$ which we found to be very good in a large parameter range (see red dashed lines in Fig. 9), and look at its scaling with the system size. In Fig. 10, we show the asymptotic values as a function of the disorder highlighting their dependence on the system size. For finite size systems it is expected that $|I|^2 > 0$ and one should therefore look at the thermodynamic limit. We extrapolate the infinite-size imbalance $I_0(V)$ from a scaling function of the form $|\mathcal{I}|^2 = I_0 + a/N$, and we observe (see inset) that it is 0, or reasonably close to it, for $V \lesssim 20$, while it increases to non-zero values for $V \gtrsim 20$, indicating a localized state where some memory of the initial state is

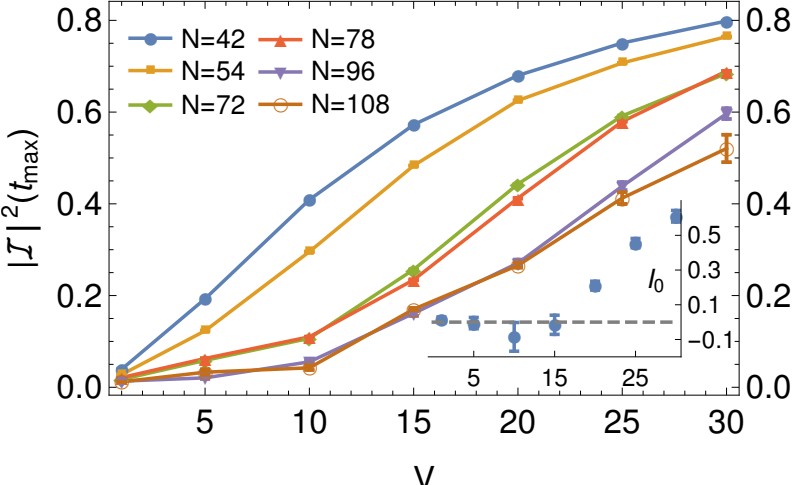

Figure 10: Asymptotic value of modulus-squared imbalance at long times, as a function of disorder. *Inset.* Coefficient $I_0$ of the scaling function $|\mathcal{I}|^2 = I_0 + a/N$ as a function of disorder.

kept at infinite time.

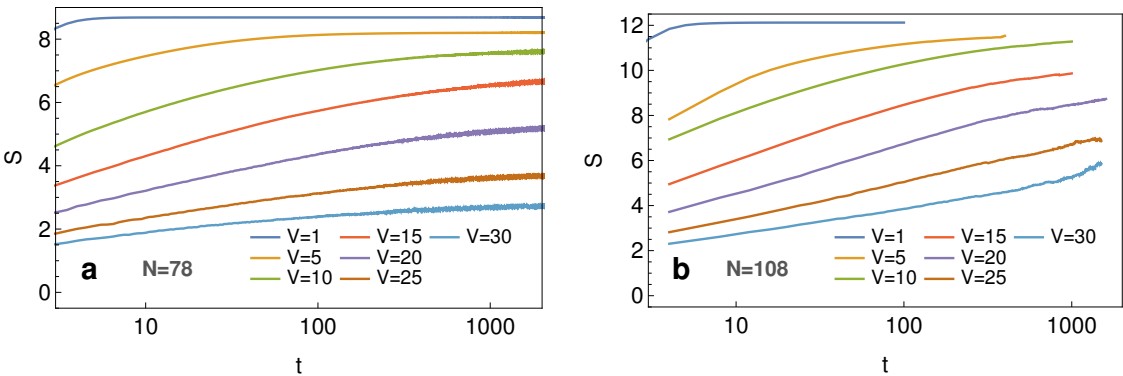

Figure 11: Bipartite entanglement entropy as a function of time for two system sizes, $N = 78$ (panel **a**) and $N = 108$ (panel **b**), showing a logarithmic growth at high disorder ($V \geq 20$).

**Entanglement entropy** A known remarkable feature of the localized phase in one dimension is a slow growth of the entanglement, which spreads logarithmically in time as opposed to a ballistic (linear in time) spread in the extended phase. In finite systems the growth is eventually limited by the corresponding volume law in the two regimes [22–24, 56–58]. We remark that given the geometry imposed by the entanglement cut of the 2D system (see Fig. 1b and 14), entanglement can spread only in the direction perpendicular to the cut, and we thus expect a spread similar to a 1D localized regime in this case.

In Fig. 11, we show the bipartite entanglement entropy, as defined in Eq. (7), as a function of time, for the cluster sizes $N = 78$ and $N = 108$. For low disorder, a fast saturation to the volume law value can be readily observed. As disorder increases, the entanglement entropy continues to quickly reach a size-dependent limiting value. For disorders $V \gtrsim 20$, a logarithmic growth appears to be present, consistent with the existence of an MBL regime. We note that this feature is only visible in the largest clusters, $N = 96$ and $N = 108$, highlighting the need of analysing very large system sizes in order to obtain evidence of a localized regime.

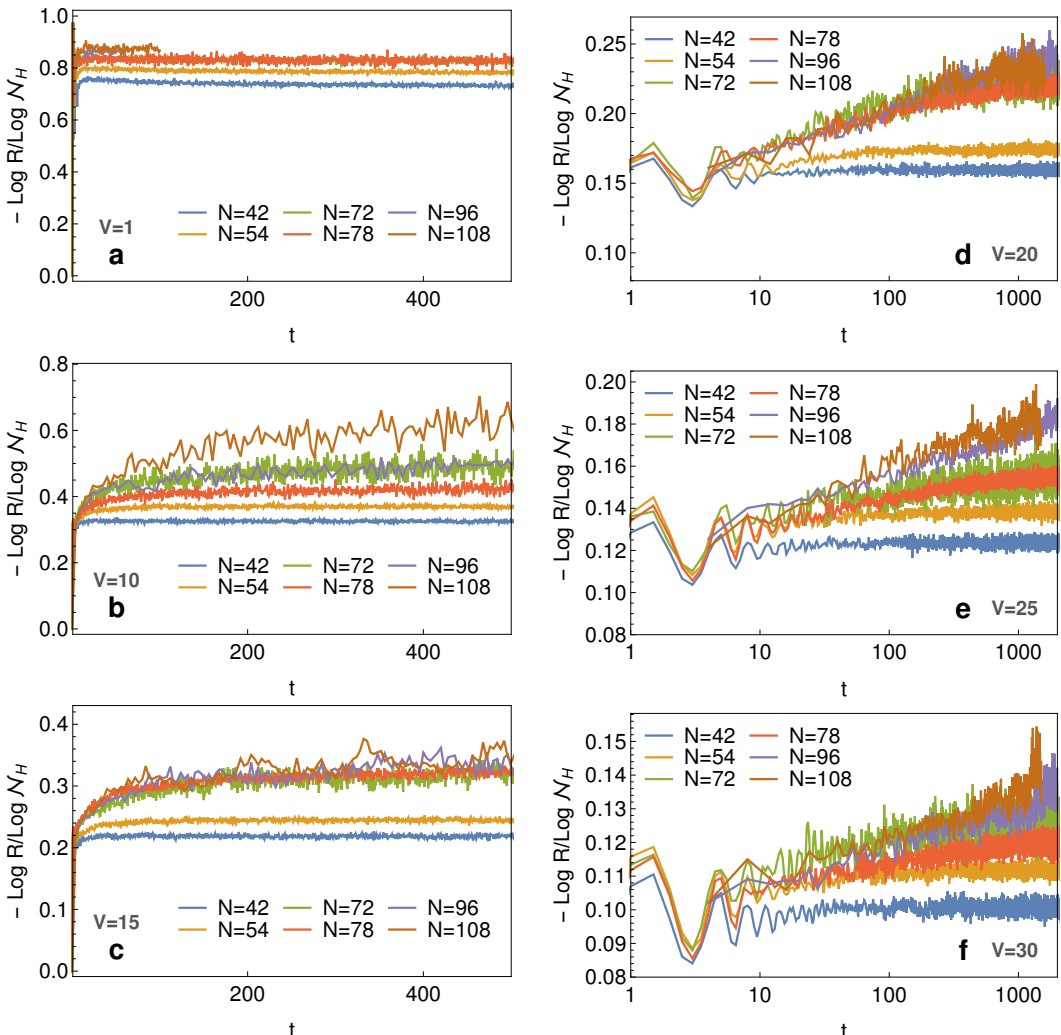

Figure 12: Time dynamics of the return probability of the initial product state, rescaled by the system size $-\ln R/\ln \mathcal{N}_H$. *Left column.* Values for low disorder: from top to bottom, $V = 1$ (panel **a**), $V = 10$ (panel **b**) and $V = 15$ (panel **c**). *Right column.* Values for high disorder, showing a logarithmic increase: from top to bottom, $V = 20$ (panel **d**), $V = 25$ (panel **e**) and $V = 30$ (panel **f**).

**Return probability** We then consider the return probability $R = |\langle \psi(t)|\psi(0)\rangle|$. Being an overlap of two vectors in the Hilbert space, one expects that it will be exponentially small (scaling as the inverse of the Hilbert space size) at long times in both ergodic and localized regimes, but its time dependence may reveal non-trivial differences. To account for the system size scaling, we consider (minus) the logarithm of the return probability rescaled with the (log of the) Hilbert space size $-\ln R/\ln \mathcal{N}_H$ which is displayed as a function of time in Fig. 12 for six values of the disorder $V$. For low disorder ($V = 1$, $V = 10$ and $V = 15$, panels **a**, **b** and **c** respectively in Fig. 12) the rescaled return probability quickly reaches a limiting value, which is smaller in absolute value as the disorder increases. At larger disorder ($V \gtrsim 20$), a logarithmic increase appears for larger system sizes, indicating a slow spreading in a range consistent with the one obtained from entanglement entropy and the participation entropy results shown below. We finally note that there is reasonable collapse between different system sizes (except the smallest two $N = 42$ and $N = 54$).

Figure 13: Participation entropy rescaled by the size of the Hilbert space $S_p/\ln\mathcal{N}_H$ as a function of time. *Left column.* Values for low disorder: from top to bottom, $V = 1$ (panel **a**), $V = 10$ (panel **b**) and $V = 15$ (panel **c**). *Right column.* Values for high disorder, showing a logarithmic growth: from top to bottom, $V = 20$ (panel **d**), $V = 25$ (panel **e**) and $V = 30$ (panel **f**).

**Participation entropy** Finally, we consider the participation entropy, as defined in Eq. (4), of the time evolved state. In Fig. 13 we show the participation entropy, rescaled by the logarithm of the Hilbert space size, as a function of time, for six values of the disorder strength. For the small disorders $V = 1$, $V = 10$ and $V = 15$, shown in panels **a**, **b** and **c** respectively, a quick saturation to system-size dependent values can be readily observed, with notably a saturation to a value very close to 1 for very small disorder; for higher, $V \geq 20$ disorders, shown in logarithmic scale in panels **d**, **e** and **f** of Fig. 4, a slow, logarithmic growth suggesting localization becomes apparent for the two largest system sizes. The behavior of the participation entropy thus closely resembles the one of the bipartite entanglement entropy.

# 4    Conclusions

The existence of a MBL transition in the thermodynamic limit in two dimensions is an important and debated topic. Given that the results presented in this manuscript rely on numerics performed on system of finite size systems, we now critically review our results in this light.

Our finite-size numerics, on systems as large as possible with unbiased numerical methods (up to $N = 78$ for exact eigenstates and $N = 108$ for dynamics) clearly distinguish two different regimes: extended at low disorder, and many-body localized at strong disorder. The later conclusion is based most noticeably on dynamical results on large sizes, which are essential to find e.g. the slow logarithmic growth of entanglement entropy characteristic of the MBL regime. Most of our data on eigenstates show a behavior that can be interpreted as a finite-size sign of a transition: the crossing point in the gap ratio and KL divergence, the pinch of curves of the participation entropy and scaled entanglement entropy. There is no real equivalent signal for dynamical data, as our disorder grid is too sparse (due to computational cost of such computations). The attempt at estimating the finite-size scaling of the infinite-time value of imbalance after a quench is nevertheless compatible with a transition too. Overall, this is very similar to what is observed for the 1D MBL transition. Our numerical results are obtained on matrix sizes similar to those used for the study of 1D spin 1/2 chains: mid-spectrum eigenstates for matrices of up to sizes $3.5 \cdot 10^5$ (comparable to the Hilbert space size for a XXZ chain with $L = 21$ spins) and dynamics on time scales corresponding to more than 1000 plaquette flips, for systems with Hilbert space size $8 \cdot 10^7$ (similar to a XXZ chain with $L = 29$ spins). The quality of the numerical results suggesting a transition to an MBL regime is of the same level of what has been obtained for one-dimensional spin chains. Similar conclusions have been reached for the disordered QDM on the square lattice [48].

However, and quite similar to the 1D case again, we believe it is currently impossible to construct an extrapolation scheme to provide a safe value for the thermodynamic value of the transition point, given the constraints in system sizes. For instance, the gap ratio or the Kullback-Leibler curves crossings display some drift, and we have too few crossing points to make a reasonable extrapolation. Extrapolating the maximum of the standard deviation of entanglement entropy to extract a thermodynamic value for a putative critical point is also difficult without any bias, as exemplified in Fig. 8. Note that this also the case for the 1D standard model of MBL [12]. We also attempted a scaling analysis (done through the bayesian method [59]) using a second-order phase transition ansatz on some of the quantities presented in Sec. 3. With the available system sizes, it was not possible to obtain a collapse. All these remarks can be possibly interpreted in two ways: (a) there is a finite value for the transition, but the finite systems considered are not large enough to be in the universal scaling regime; (b) there is no transition in the thermodynamic limit, but only a crossover to increasingly slow dynamics.

We furthermore point out that an attempt at extrapolation also suffers from the absence of an established finite-size scaling theory for MBL transitions. In one dimension, earlier real-space renormalization group suggested a continuous transition [60, 61] while more recent works suggest a Kosterlitz-Thouless (KT) transition [62, 63] or a KT-like transition albeit with different scaling [64]. In any case, these theories only consider the 1D MBL transition and do not apply to a putative 2D MBL case. Furthermore, as emphasized in previous work on the square lattice [48], constrained models do not easily fit in the local integrals of motions picture, the basis for most analysis of MBL and MBL transitions.

We conclude by remarking that alternative opportunities for testing the stability of the MBL regime on larger systems come from experimental realizations in specifically arranged experimental setups. Our work highlights the interest in studying constrained models. There has been a lot of recent effort devoted to perform analog quantum simulations of lattice gauge

theories (see Ref. [65] for a recent review), in order to implement experimentally e.g. the Gauss law equivalent to the dimer constraint. Let us for instance mention explicit proposals for implementing QDMs with different possible setups using Rydberg atoms [66–68]. Even if the above second scenario (b) is ultimately true for the disordered QDM on the honeycomb lattice, our results indicate that the time scales for thermalization are very long, meaning that the system will be effectively localized at high disorder for all practical purposes in these potential experimental platforms.

Besides the possibility of many-body localization, it would be interesting to see whether the constraints and the non-tensor product structure in QDM could allow the existence of quantum scar states [69], similar e.g. to what happens in the 1D constrained PXP model. These scar states have been argued to realize intermediate scenarios between the extended and localized paradigms. A recent work has indeed identified quantum scars on a specific QDM on a non-bipartite kagome lattice [70].

## Acknowledgments

We thank Jean-Marie Stéphan for an useful exchange. This work benefited from the support of the project THERMOLOC ANR-16-CE30-0023-02 of the French National Research Agency (ANR) and by the French Programme Investissements d'Avenir under the program ANR-11-IDEX-0002-02, reference ANR-10-LABX-0037-NEXT. This project has received funding from the European Union's Horizon 2020 research and innovation programme under the Marie Sklodowska-Curie grant agreement No 838773. We acknowledge PRACE for awarding access to HLRS's Hazel Hen computer based in Stuttgart, Germany under grant number 2016153659, as well as the use of HPC resources from CALMIP (grants 2017-P0677 and 2018-P0677) and GENCI (grant x2018050225). The computer codes that allowed to obtain the results presented in Sec. 3 make use of the libraries PETSc [71,72], SLEPc [73,74] and Strumpack [75,76].

## A   Overview of the finite-size lattices used

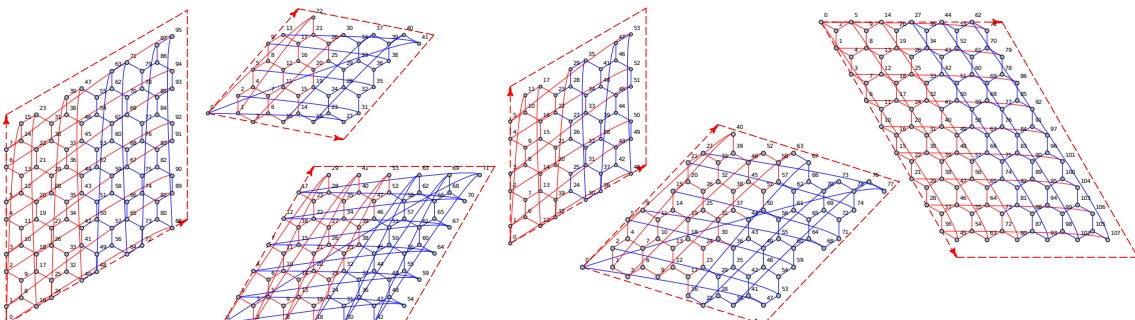

Figure 14: Overview of the honeycomb lattice clusters, with periodic boundary conditions, that have been used for exact diagonalization and dynamics. The red and blue subsystems correspond to the ones used for entanglement entropy computations.

In this work we have used the honeycomb lattices with $N = 42$, 54, 72, 78, 96 and 108 sites shown in Fig. 14. These were all considered with periodic boundary conditions and are constructed with the following basis vectors [77], written in the basis $\{u_1, u_2\}$ where $u_1 = (1, 0)$ and $u_2 = (1/2, \sqrt{3}/2)$:

The separation into two subsystems used for the calculation of the bipartite entanglement

Table 2: Vectors defining the honeycomb lattice clusters that have been used, written in the basis $\{u_1, u_2\}$ where $u_1 = (1, 0)$ and $u_2 = (1/2, \sqrt{3}/2)$.

| $N$ | $v_1$ | | $v_2$ | |
|---|---|---|---|---|
| 42 | (1 | 4) | (5 | −1) |
| 54 | (3 | 3) | (6 | −3) |
| 72 | (6 | 0) | (6 | −6) |
| 78 | (2 | 5) | (7 | −2) |
| 96 | (4 | −8) | (4 | −4) |
| 108 | (6 | 0) | (9 | −9) |

entropy is shown in different colors in each cluster. The boundary has been chosen parallel to one of the basis vectors. We note that in some cases (namely, clusters $N = 54$ and $N = 78$) this was not exactly possible but was chosen as close as possible to the parallel boundary line.

## B  Mobility edge

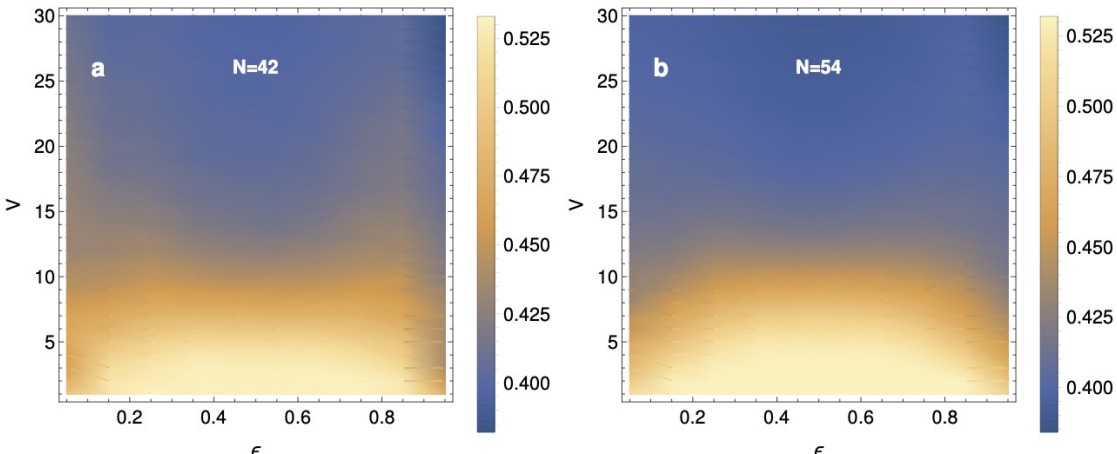

Figure 15: Color map plot of the gap ratio $r$ as a function of the disorder strength $V$ and the rescaled energy $\epsilon$ for the $N = 42$ (panel **a**) and $N = 54$ (panel **b**) cluster sizes, showing an energy-dependent mobility edge.

We present here an additional analysis of the gap ratio defined in Sec. 3.1, this time resolved in energy. The purpose is to identify a possible dependence of the localization transition value from the energy, i.e. the presence of a so-called mobility edge [12].

For the smallest system sizes, we consider full exact diagonalization. As customary, we introduce the parameter

$$\epsilon = \frac{E - E_{\min}}{E_{\max} - E_{\min}} \in [0, 1]. \tag{9}$$

Thus, from the whole spectrum, we compute the gap ratio for ten $\epsilon$ windows of fixed width and average on around 1000 disorder realizations. The result for cluster sizes $N = 42$ and $N = 54$ is shown in Fig. 15. Having only the two smallest system sizes available, we cannot definitively conclude the existence of a mobility edge in the model (1), although Fig. 15 does show an indication of an enhanced localization at the spectrum extrema, which appears more marked for $N = 54$ than for $N = 42$.

## C  Area law entanglement of selected states

Given the different symmetries and aspect ratios of the clusters, dividing them in two subsystems for the purpose of the computation of the bipartite entanglement entropy should be done respecting the vectors of each cluster, as outlined in Appendix A. In order to check that the chosen cut is sufficiently general, we computed the entanglement entropy of some reference states which are known to have an area law as the system size increases. The entanglement entropy, rescaled by the area of the cut, is shown in Fig. 16. The reference states are: the ground state $|\psi_{GS}\rangle$ of the nondisordered model with constant potential $V_c = 0.1$; the 'Rokshar-Kivelson' [45] (RK) state, defined as the vector $|\psi_{RK}\rangle = 1/\sqrt{\mathcal{N}_H}|1\,1\ldots 1\rangle$, where the elements have equal amplitude in the dimer covering configurations basis; two localized states at high disorder, respectively obtained at disorder strength $V = 30$ and $V = 50$. For all states, $S/\mathcal{A}$ is approximately constant with respect to system size $N$, showing thus the correct area law scaling for the selected cut in all the clusters shown in Fig. 14. For the RK state, we consider only configurations in the $(0,0)$ winding sector to allow a comparison with other states. The entanglement entropy of the RK state can rigorously be shown to sustain an area law [78], this independently of whether one considers all states or only states in a fixed winding sector.

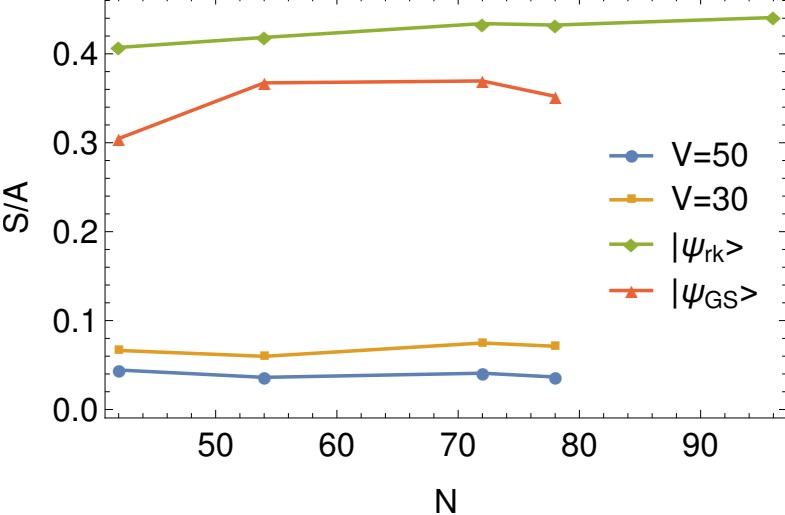

Figure 16: Bipartite entanglement entropy of some reference states in the honeycomb lattice rescaled by the size of the boundary between the two subsystems, $S/\mathcal{A}$, showing an area law scaling. Shown are entanglement scalings for the ground state $|\psi_{GS}\rangle$ of the non-disordered model with constant field $V_c = 0.1$, the uniform state $|\psi_{RK}\rangle$, and the localized states at disorder $V = 30$ and $V = 50$.

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
