# Peer review of "Probing many-body localization in a disordered quantum dimer model on the honeycomb lattice"

_SciPost Physics, doi:SciPost Phys. 10, 044 (2021)_

## Round 2 · Referee Report · Elmer Doggen (Referee 1) · 2020-7-3

Strengths

1-The authors consider a model that (to my knowledge) has not been considered before in the context of many-body localization (MBL), though bearing similarities to the previous study Ref. [48].
2-The authors use many different measures as a probe for MBL.
3-Relatively few studies into 2-dimensional MBL have been performed so far, the work is timely and relevant.
4-The findings are relevant to state-of-the-art experiments.

Weaknesses

1-The authors do not clearly demonstrate that MBL properties survive in the thermodynamic limit - although it is not necessarily expected that this would be the case.

Report

This work addresses the problem of many-body localization (MBL), which pertains to the interplay between disorder and interactions in many-body systems. At this point many numerical works have been devoted to MBL in one-dimensional systems, particularly in the archetypal model: the XXZ Heisenberg chain. There are strong indications that in this system there is a transition in the thermodynamic limit between an ergodic and a localized phase at a certain finite (model-dependent) strength of the disorder.

Of major interest is what happens in the case of two geometric dimensions (2D). Some theoretical approaches and recent experiments indicate that a "true" transition remains in this case, whereas some theoretical methods and our own recent numerics (Ref. [39]) suggest instead that in 2D the "transition" is more like a crossover, with an effective critical disorder that grows with system size.

The challenge for numerical approaches in 2D is that in even the simplest unconstrained many-body systems the Hilbert space growth exponentially in the number of sites N as 2N. Since exact numerics ("unbiased" in the words of the authors) can handle up to 20-ish sites, this leaves the available system sizes prohibitively small. A way to circumvent this problem is by introducing constraints in the Hamiltonian, which reduces the scaling of the size of the Hilbert space with the number of sites.

The present manuscript considers such an approach in a constrained dimer model on a honeycomb lattice. The authors then proceed to apply a large distinct number of measures of the MBL transition to the model, including both static properties such as level statistics and entanglement, as well as dynamics. They argue that their results indicate an MBL transition in this model at a disorder value of V20-25.

This work represents a nice addition to the existing literature on MBL, in particular adding to the results in two dimensions where relatively few results are available, providing high-quality numerical results. In my view there is still a lot of potential for understanding MBL through the pathway of similar constrained models. I would recommend publication in SciPost Physics as the manuscript successfully addresses Expectation criteria 1 and 3.

I have the following comments/questions:

1-I am confused concerning the discussion around Fig. 5. How are the black dashed lines computed here? What is their precise relation to the basis states?

2-I assume that the way the system is "sliced" for the bipartite entropy is not crucial for the results. Did the authors explicitly verify this?

3-A key question concerns the extrapolation of the results to large systems. The curves for the imbalance dynamics in Fig. 9 show quite a strong dependence on system size, seemingly much stronger than similar curves for the imbalance in 1D. I wonder if a crossover-type scenario along the lines of [Phys. Rev. B 99, 134305] might be a better explanation of the authors' results for this reason - although one should always be careful extrapolating from small systems of course. The authors fairly acknowledge such a possibility in the Conclusion, but perhaps it would be useful to explore this scenario in more detail.

Requested changes

1-The axis labels are often tiny and hard to read. Please update the figures and make them more readable, using fewer columns if needed. Also, if there is a left and right panel, explain the left panel first (Fig. 2). If a figure has many colours, I recommend using a gradient (e.g. equally spaced colours from a "viridis" colourmap).

2-I'm not a big fan of this "biased/unbiased"-nomenclature, but if you must use it, please define clearly what is meant by this (exact vs. approximate approaches).

---

## Round 2 · Referee Report · Anonymous (Referee 3) · 2020-7-22

Strengths

1- a disordered quantum dimer model on a honey comb lattice is studied, which is an open problem
2- state-of-the-art exact diagonalization is used to investigate the problem

Weaknesses

1- the presented data does not sufficiently support the claim that an MBL transition exists in this system --- different observables show different behavior and finite size effects are strong.

Report

The authors study a disordered quantum dimer model on the honey comb lattice using exact diagonalization approaches (full diagonalization, shift and invert techniques, and Lanczos time evolution). The goal is to identify whether the system possesses a many-body localization (MBL) transition at some critical value of the disorder. This is a relevant question, because it is a subject of debate whether such a transition can exist in dimensions greater than one. The authors provide a comprehensive numerical study of the problem, by both looking at spectral statistics and non-equilibrium dynamics.

When studying the data, it seems to me that there is no clear evidence for an MBL transition. Here are a couple of examples from which one would rather expect the absence of a transition in the thermodynamic limit:
1- Fig. 1: the level spacing statistics shows significant finite size effects in the crossing point of different system sizes
2- Fig. 8: the peak in the variance of the entanglement entropy is significantly shifting to larger values of the disorder strength when the system size is increased
3- Fig 9: The dynamics of the imbalance is still strongly finite size dependent with a trend that actually at larger system sizes there is stronger decay (even though times are not as large as in panel a)

I think that the authors need to critically discuss these issues when revising the manuscript. Also another challenge (which is presented somewhat differently in the manuscript) is indeed the slow growth of the Hilbert space with system size N, because in that case the argument for destabilization of many-body localization provided in Phys. Rev. B 95, 155129 (2017), will be even weaker. Due to these considerations it seems to be very challenging to affirmatively answer whether there exists an MBL transition in this 2D quantum dimer model. It would be great if the authors could discuss this point too.

Requested changes

1- critically discuss possible discrepancies between the different investigated observables, in particular, also focussing on finite size effects.
2- Pg. 5, 2nd paragraph: rewrite the terms localized phase and ETH phase.
3- Fig. 3: Maybe it makes sense to show the KL divergence divided by its system size
4- Fig. 10 is misleading. It is not justified to take finite time date (on top of that at different times) and make an attempt to extrapolate them to infinite system size --- in particular given the residual flow of the imbalance for the larger system at strong disorder. Please remove this figure.
5- On page 11 top and the appendix the description of the RK point is misleading as one would think that it is attained at V=0 (it is characterized as model 1 in the absence of disorder). Also what is meant with the ket of ones (all possible dimer coverings in the zero winding sector, I suppose?) should be clarified.
6- The authors state that the RK point has area law scaling of the entanglement entropy. This statement is clear when considering all winding sectors. When projecting onto winding zero it is not obvious to me because of the entropy which results from fixing the winding constraint. Please elaborate on that point.
7- Sometimes in the text the figures are referred to as top and bottom panel instead of left and right panel. I suggest to introduce a) and b) labels for clarity.

---

## Round 3 · Referee Report · Elmer Doggen (Referee 1) · 2021-1-4

Report

The authors have adequately responded to the referees' comments. I recommend the manuscript for publication, but I have just one more small question concerning the transient $1/\sqrt{t}$-behaviour. Is this just a phenomenological fit or is it justified in some way? In the case of standard MBL in $D$ dimensions, we expect diffusive behaviour $\propto 1/t^{D/2}$ in the weakly disordered case, but that probably does not apply to this constrained $2D$ model at stronger disorder.
  • validity: high
  • significance: good
  • originality: good
  • clarity: high
  • formatting: good
  • grammar: good

Author:  Francesca Pietracaprina  on 2021-01-04  [id 1124]

(in reply to Report 1 by Elmer Doggen on 2021-01-04)
Category:
answer to question

We thank the referee once again for their work and for their recommendation.
Regarding the last question: we have no simple explanation for the $1/\sqrt{t}$ fit which is phenomenological, albeit of excellent quality in a large time window. We will explicitly add the fact that the fit is phenomenological in the final version of the manuscript.

---

## Round 3 · Referee Report · Anonymous (Referee 2) · 2021-2-1

Report

The authors have clarified my questions and comments in the revised version of the manuscript, which hence can be recommended for publication.

---

## Round 3 · Author Response

Dear Editor,

we thank you for arranging the review of our manuscript and we are grateful to the referees for the time they invested in reading it as well as for their careful reports and constructive comments. We have revised and improved our manuscript taking into account the remarks that have been raised and we submit here the updated version. Below, we reply point-by-point to each of the comments and describe in detail the modifications that have been made.

We hope that with these improvements our manuscript can be readily considered for publication in Scipost physics.

Yours sincerely,

The authors

First report (Dr. Elmer Doggen)

This work addresses the problem of many-body localization (MBL), which pertains to the interplay between disorder and interactions in many-body systems. At this point many numerical works have been devoted to MBL in one-dimensional systems, particularly in the archetypal model: the XXZ Heisenberg chain. There are strong indications that in this system there is a transition in the thermodynamic limit between an ergodic and a localized phase at a certain finite (model-dependent) strength of the disorder.

Of major interest is what happens in the case of two geometric dimensions (2D). Some theoretical approaches and recent experiments indicate that a "true" transition remains in this case, whereas some theoretical methods and our own recent numerics (Ref. [39]) suggest instead that in 2D the "transition" is more like a crossover, with an effective critical disorder that grows with system size.

The challenge for numerical approaches in 2D is that in even the simplest unconstrained many-body systems the Hilbert space growth exponentially in the number of sites N as 2^N. Since exact numerics ("unbiased" in the words of the authors) can handle up to 20-ish sites, this leaves the available system sizes prohibitively small. A way to circumvent this problem is by introducing constraints in the Hamiltonian, which reduces the scaling of the size of the Hilbert space with the number of sites.

The present manuscript considers such an approach in a constrained dimer model on a honeycomb lattice. The authors then proceed to apply a large distinct number of measures of the MBL transition to the model, including both static properties such as level statistics and entanglement, as well as dynamics. They argue that their results indicate an MBL transition in this model at a disorder value of V ≈ 20-25.

This work represents a nice addition to the existing literature on MBL, in particular adding to the results in two dimensions where relatively few results are available, providing high-quality numerical results. In my view there is still a lot of potential for understanding MBL through the pathway of similar constrained models. I would recommend publication in SciPost Physics as the manuscript successfully addresses Expectation criteria 1 and 3.

I have the following comments/questions:

  • I am confused concerning the discussion around Fig. 5. How are the black dashed lines computed here? What is their precise relation to the basis states?

Reply:

We thank the referee for highlighting this possibly confusing point.

The imbalance, as defined in Eq. 4, can be computed directly for the basis states (i.e. the dimer configurations of the honeycomb lattice). This corresponds to the infinite-disorder limit in the Hamiltonian H_{QDM}. In Fig. 5, the black dashed line shows the probability distribution of the imbalance for these basis states (which are eigenstates at infinite disorder). The peaks correspond to each basis state, and can then be compared with the finite-disorder results shown as colored lines: at sufficiently high disorder, peaks start to develop in the same positions, showing an imbalance increasingly close to the basis states.

We have clarified the meaning of the black lines of Fig. 5 in the main text.

  • I assume that the way the system is "sliced" for the bipartite entropy is not crucial for the results. Did the authors explicitly verify this?

Reply:

This is indeed a reasonable concern and indeed some special ways of cutting the honeycomb clusters could in principle have peculiar scaling and not the general properties that we are interested in. The cut we consider (approximately parallel to the boundary) is the most natural one with the volumes (of A and B) scaling as L^2 and the area scaling as L.

We have taken steps in order to verify that the cut we used was sufficiently general. We did this by testing for an area law and volume law for the chosen bipartition cut. Namely, in appendix C we verified the area law scaling of the entanglement on some well known cases: the ground state of the non-disordered model with constant potential V_e = 0.1 and the `Rokshar-Kivelson' state $1/\sqrt{\mathcal N_H} |{1\,1\,\dots\,1}\rangle$. These states indeed display an area law scaling as expected, which confirms the well-fondness of the chosen cut. At the same time, in these cluster, for random states, the entanglement entropy reaches a value which scales, as expected, as the volume of the cluster (this is shown in the dashed lines in Fig. 7).

  • A key question concerns the extrapolation of the results to large systems. The curves for the imbalance dynamics in Fig. 9 show quite a strong dependence on system size, seemingly much stronger than similar curves for the imbalance in 1D. I wonder if a crossover-type scenario along the lines of [Phys. Rev. B 99, 134305] might be a better explanation of the authors' results for this reason - although one should always be careful extrapolating from small systems of course. The authors fairly acknowledge such a possibility in the Conclusion, but perhaps it would be useful to explore this scenario in more detail.

Reply:

We thank the referee for this comment, which is similar in spirit to the main criticism of the second referee.

We highlight here that indeed distinguishing between a crossover to an extremely slow dynamics and a phase transition to a localized system at finite disorder strength is an especially hard task to resolve definitively using numerics for (relatively) small system sizes. One of the arguments for a crossover would be the difficulty of doing rigorous finite size scaling. %This is highlighted in the scaling of the peak of the entanglement entropy variance (see modified Figure 8), where it is virtually impossible to extrapolate to the thermodynamic limit without further assumption. A good point in favor of the existence of true localized regime is the localized phenomenology that emerges for all the quantities and especially in the dynamics results for the largest system sizes N = 108 and N = 96. These two systems sizes show, for example, a logarithmic growth of entanglement which is one of the most peculiar and recognizable features of many-body localization; we also note that this is invisible in smaller sizes. Thus, it becomes especially necessary to reach very large system sizes to see a truly localized phenomenology.

With respect to the mentioned strong dependence on system size for the time-asymptotic value of the imbalance (shown in Fig. 9 and 10), we did attempt to estimate the infinite-size value (now with a t^(-1/2) fit, which we found to be very good in a large parameter range, see Fig. 9): from our results, it seems that a scaling function of the form ~ a/N describes the finite-size effects well.

Overall, given these numerical indications, we claim that the evidence supporting many-body localization in the constrained model under analysis is of similar quality to the numerical evidence supporting MBL in 1D spin chains. Nevertheless, we recognize that this numerical evidence cannot be a definitive proof supporting the existence of either a transition or a crossover, especially given the delicate nature of system size extrapolation in these systems. We outline this in a critical discussion in the Conclusion section (now rewritten and extended in the new manuscript), as well as in attempts at extrapolating the value of the entanglement variance maximum (adding a discussion in Sec. 3.3 and a new panel in Fig. 8). We believe that our numerical results (once again at the state-of-the-art for exact diagonalization of large systems) are in any case a strong building block for testing the existence of 2D MBL, possibly with experimental methods.

Requested changes

  • The axis labels are often tiny and hard to read. Please update the figures and make them more readable, using fewer columns if needed. Also, if there is a left and right panel, explain the left panel first (Fig. 2). If a figure has many colours, I recommend using a gradient (e.g. equally spaced colours from a "viridis" colourmap).

Reply:

We thank the referee for this useful comment about the clarity of our figures. We have increased the size of some figures in order to increase their legibility and changed the colors, namely in figures 2 and 6.

  • I'm not a big fan of this "biased/unbiased"-nomenclature, but if you must use it, please define clearly what is meant by this (exact vs. approximate approaches).

Reply:

We understand the referee's criticism of the nomenclature that we use to describe the exact diagonalization method (as opposed to other methods which may involve approximations). We have left this nomenclature but we have taken care in defining it in the Introduction to avoid any misunderstanding of what we mean by ``unbiased'' in the present context.

Second report

The authors study a disordered quantum dimer model on the honey comb lattice using exact diagonalization approaches (full diagonalization, shift and invert techniques, and Lanczos time evolution). The goal is to identify whether the system possesses a many-body localization (MBL) transition at some critical value of the disorder. This is a relevant question, because it is a subject of debate whether such a transition can exist in dimensions greater than one. The authors provide a comprehensive numerical study of the problem, by both looking at spectral statistics and non-equilibrium dynamics.

When studying the data, it seems to me that there is no clear evidence for an MBL transition. Here are a couple of examples from which one would rather expect the absence of a transition in the thermodynamic limit:

  • Fig. 1: the level spacing statistics shows significant finite size effects in the crossing point of different system sizes
  • Fig. 8: the peak in the variance of the entanglement entropy is significantly shifting to larger values of the disorder strength when the system size is increased
  • Fig 9: The dynamics of the imbalance is still strongly finite size dependent with a trend that actually at larger system sizes there is stronger decay (even though times are not as large as in panel a)

I think that the authors need to critically discuss these issues when revising the manuscript.

Reply:

We thank the referee for their comment. We definitely agree that definitively concluding the existence of an MBL transition is a hard claim to make in general with numerical methods. For the level spacing statistic and the peak in the the entanglement entropy variance, we agree that there is a significant drift for the system sizes shown. We refer to our answer to the third comment of the first referee, which better explains this point and the dynamics of the imbalance (see also below). On the other hand, this is all very similar to what happens in the 1D XXZ spin chain, for which MBL is quite strongly established. Our conclusions overall, based on a detailed analysis, is that there are clearly two extended and MBL regimes, separated by a transition. However it is not possible to extrapolate the value of the transition point to the thermodynamic limit, given the system sizes considered and the absence of a finite-size scaling theory for MBL transition.

We have modified the conclusions, in order to provide an extended critical discussion of our results and the possibility of a MBL (transition or phase) in 2D.

With respect to the finite size effects in the imbalance dynamics, which are indeed present in Fig. 9, we remark that the asymptotic value (estimated from the data in Fig. 9 and shown as a function of the disorder in Fig. 10) is actually subject to a relatively good control, with a good fit to a 1/t^(1/2) decay which is now displayed in the new manuscript. Consequently, we are able to obtain a reasonable estimate of the infinite-time value, for which we then attempt a finite-size scaling. The resulting infinite-size asymptotic imbalance I_0 (see inset of Fig. 10) displays sign of a transition at around V ≈ 20. Please also see our reply to the referee's comment on Fig. 10 below.

Also another challenge (which is presented somewhat differently in the manuscript) is indeed the slow growth of the Hilbert space with system size N, because in that case the argument for destabilization of many-body localization provided in Phys. Rev. B 95, 155129 (2017), will be even weaker. Due to these considerations it seems to be very challenging to affirmatively answer whether there exists an MBL transition in this 2D quantum dimer model. It would be great if the authors could discuss this point too.

Reply:

While it is true that the thermalization phenomenon must be slower (for a given number of sites) since the many-body gaps close more slowly in the middle of the spectrum (as 1.15^N instead of 2^N for a spin 1/2 chain), we can reach larger sample sizes N. However what really matters for the thermalization-ETH arguments are actually the maximal matrix sizes that we consider (and the corresponding level spacing). We obtain mid-spectrum eigenstates for matrices of up to sizes 3.5 ⋅ 10^5 (comparable to the Hilbert space size for a spin 1/2 XXZ chain with L = 21 spins, close to the state of the art) -- we couldn't go to the next available matrix size (6 ⋅ 10^6 for the next available sample N = 96). Furthermore, we also obtain dynamics on time scales corresponding to more than 1000 plaquette flips, for systems with Hilbert space size ~8 ⋅ 10^7 (similar to a a spin 1/2 XXZ chain with L = 29 spins, which is the state of the art). We thus believe that the numerical results presented here are of comparable quality, and thus should be able to reach similar conclusions, as for 1D systems. Of course this doesn't ensure that they can be considered definitive (see discussion above and in the conclusions) most notably on the existence of a transition in the thermodynamic limit.

Requested changes

  • critically discuss possible discrepancies between the different investigated observables, in particular, also focussing on finite size effects.

Reply:

We now adress this point in detail in the new discussion part included in the conclusions.

  • Pg. 5, 2nd paragraph: rewrite the terms localized phase and ETH phase.

Reply:

Since we agree that the wording phase'' should be reserved to thermodynamic behavior, we have changed in most cases to the wordingETH'' and ''localized`` regimes, in particular in the paragraph mentionned by the Referee.

  • Fig. 3: Maybe it makes sense to show the KL divergence divided by its system size

Reply:

We have added an extra panel in Fig. 3 in support of the expected scaling of KL in the localized regime stated in the main text.

  • Fig. 10 is misleading. It is not justified to take finite time date (on top of that at different times) and make an attempt to extrapolate them to infinite system size --- in particular given the residual flow of the imbalance for the larger system at strong disorder. Please remove this figure.

Reply:

We kindly disagree with the referee with respect to the validity of Fig. 10. Indeed, for system sizes 42 and 54, we performed full ED and we were able to obtain the asymptotic imbalance. For larger sizes, we used Krylov subspace methods for time evolution, which does limit, computationally, the maximum time reached. However, we carefully considered large enough time to ensure that saturation had been reached (i.e. t = 5 ⋅ 10^3 for sizes 72 and 78); an example is the data shown in Fig. 9 (for N = 78), where the residual flow is arguably absent. To highlight this, we now show a fit of the simple form $a/\sqrt{t}+c$, which is very good in a large time window. The asymptotic value obtained from the fit is shown in black in Fig. 9. For the largest system sizes, we were more constrained with respect to the maximum time and some residual flow is indeed visible for N = 108 (most evidently at V = 5 and V = 30 but also to a lesser degree for the other values of V considered). We now consider the fit result for the asymptotic imbalance value for the points in fig. 10. This helps avoid the systematic overestimation that could have resulted from considering only the finite time data. We have modified the figures and the description of the |I|^2(t_{max}) computation in the text to reflect these changes; we note that the end result is not modified, showing a non-zero I_0 at values V ≳ 20. We thus believe that the plot in Fig. 10 is significant and we would disagree to the suggestion of removing it.

  • On page 11 top and the appendix the description of the RK point is misleading as one would think that it is attained at V = 0 (it is characterized as model 1 in the absence of disorder). Also what is meant with the ket of ones (all possible dimer coverings in the zero winding sector, I suppose?) should be clarified.

Reply:

We thank the referee for pointing out this possibly confusing wording. We define the RK state for the honeycomb lattice as $\psi_{\rm RK}=1/\sqrt{{\mathcal N}_H} \ket{1\,1\,\dots\,1}$ that is the vector for which every basis element (in the zero winding sector) has equal amplitude. We have reworded the definition.

  • The authors state that the RK point has area law scaling of the entanglement entropy. This statement is clear when considering all winding sectors. When projecting onto winding zero it is not obvious to me because of the entropy which results from fixing the winding constraint. Please elaborate on that point.

Reply:

RK points enjoy the property that their entanglement entropy can be expressed as a sum over ``classical configurations'' of their boundaries, $S = - \sum_c p_c \log(p_c)$ , where p_c is the probability of the boundary configuration. This has been first shown in Stéphan et al., Phys. Rev. B 80, 184421 (2009). The sum over c is over 2^L boundary configurations for a boundary scaling with L. The RK point thus follows an area law as its entanglement entropy is at most L log(2). For a fixed winding sector, the same construction (sum over classical boundary conditions) works, and the fixed winding entanglement entropy is similarly bounded by L log(2), hence it also follows an area law. This is indeed what we observe in the numerics in the Appendix. We added a sentence explaning this, as well as the reference to Stéphan et al.'s work in the new version of the manuscript.

  • Sometimes in the text the figures are referred to as top and bottom panel instead of left and right panel. I suggest to introduce a) and b) labels for clarity.

Reply:

We thank the referee for raising this possible confusion in the figure labelling: we have added labels in all multiple-panel figures.

---

## Round 3 · List of Changes

• We have clarified the meaning of the black lines of Fig. 5 in the main text and in the figure.
  • We have rewritten and extended the Conclusion section.
  • We added a discussion about extrapolating the value of the entanglement variance maximum in Sec. 3.3 and a new panel in Fig. 8.
  • We have increased the size of some figures in order to increase their legibility and changed the colors, namely in figures 2 and 6.
  • We have clarified the definition of "unbiased method" in the Introduction.
  • We now display the 1/t^(1/2) fit of the imbalance in Fig. 9 and comment on the fit in the main text and in the figure's caption.
  • We have changed in all appropriate cases the wording for ETH and localized "phases" to "regimes".
  • We have added an extra panel in Fig. 3.
  • We now consider the fit result for the asymptotic imbalance value (shown in Fig. 9) for the points in fig. 10.
  • We have reworded the definition of RK state.
  • We added a sentence explaning RK's entanglement entropy properties in the Appendix and a reference to Stéphan et al. Phys. Rev. B 80, 184421 (2009)
  • We have added labels in all multiple-panel figures.

---

## Editorial Decision

published